# Effects of Incorporating Ionic Crosslinking on 3D Printing of Biomass–Fungi Composite Materials

**DOI:** 10.3390/biomimetics9070411

**Published:** 2024-07-06

**Authors:** Al Mazedur Rahman, Yeasir Mohammad Akib, Caleb Oliver Bedsole, Zhijian Pei, Brian D. Shaw, Chukwuzubelu Okenwa Ufodike, Elena Castell-Perez

**Affiliations:** 1Department of Industrial & Systems Engineering, Texas A&M University, College Station, TX 77843, USA; almazedurrahman@tamu.edu (A.M.R.); yeasir.akib@tamu.edu (Y.M.A.); 2Department of Plant Pathology and Microbiology, Texas A&M University, College Station, TX 77845, USA; olib@tamu.edu (C.O.B.); brian.shaw@ag.tamu.edu (B.D.S.); 3Department of Engineering Technology and Industrial Distribution, Texas A&M University, College Station, TX 77843, USA; ufodike@tamu.edu; 4Department of Mechanical Engineering, Texas A&M University, College Station, TX 77843, USA; 5Department of Biological and Agricultural Engineering, Texas A&M University, College Station, TX 77843, USA; ecastell@tamu.edu

**Keywords:** 3D printing, ionic crosslinking, fungal growth, sodium alginate, biomass–fungi

## Abstract

Biomass–fungi composite materials primarily consist of biomass particles (sourced from agricultural residues) and a network of fungal hyphae that bind the biomass particles together. These materials have potential applications across diverse industries, such as packaging, furniture, and construction. 3D printing offers a new approach to manufacturing parts using biomass–fungi composite materials, as an alternative to traditional molding-based methods. However, there are challenges in producing parts with desired quality (for example, geometric accuracy after printing and height shrinkage several days after printing) by using 3D printing-based methods. This paper introduces an innovative approach to enhance part quality by incorporating ionic crosslinking into the 3D printing-based methods. While ionic crosslinking has been explored in hydrogel-based bioprinting, its application in biomass–fungi composite materials has not been reported. Using sodium alginate (SA) as the hydrogel and calcium chloride as the crosslinking agent, this paper investigates their effects on quality (geometric accuracy and height shrinkage) of 3D printed samples and physiochemical characteristics (rheological, chemical, and texture properties) of biomass–fungi composite materials. Results show that increasing SA concentration led to significant improvements in both geometric accuracy and height shrinkage of 3D printed samples. Moreover, crosslinking exposure significantly enhanced hardness of the biomass–fungi mixture samples prepared for texture profile analysis, while the inclusion of SA notably improved cohesiveness and springiness of the biomass–fungi mixture samples. Furthermore, Fourier transform infrared spectroscopy confirms the occurrence of ionic crosslinking within 3D printed samples. Results from this study can be used as a reference for developing new biomass–fungi mixtures for 3D printing in the future.

## 1. Introduction

Biomass–fungi composite materials comprise two primary constituents: biomass particles sourced from agricultural residues (such as corn stover, beechwood sawdust, and hemp hurd) and a network of fungal hyphae binding the biomass particles [1]. These composite materials can be used to fabricate products traditionally manufactured from petroleum-based plastics. Their potential applications span several industries, including packaging [2,3,4,5], furniture [6], and construction [7]. They offer great thermal and acoustic insulation properties [8,9,10] and are relatively inexpensive, with raw material costs ranging from USD 0.07 to USD 0.17 per kilogram [9]. In contrast, for plastic materials, raw material cost ranges from USD 2.1 to USD 2.3 per kilogram for polystyrene, from USD 1.7 to USD 1.9 per kilogram for phenolic formaldehyde resin, and from USD 8.2 to USD 10.2 per kilogram for polyurethane [9]. Furthermore, upon reaching the end of their service life, these composite materials biodegrade much faster than plastic waste, which often persists in landfills for years [11].

3D printing-based methods for manufacturing products using biomass–fungi composite materials offer an alternative to traditional molding-based methods. 3D printing enables manufacturing parts with complex shapes in art, architecture, interior design, packaging, and construction [12,13,14,15,16,17] that cannot be easily produced using molding-based methods. The processes of one 3D printing-based manufacturing method are shown in Figure 1 and described below.

Recently, several papers presented studies on 3D printing of biomass–fungi composite materials. One paper [18] showed that fungi could survive the mixing and printing processes and continue growing on the printed samples. A subsequent paper from the same authors [12] reported the effects of mixture composition on rheological properties of biomass–fungi mixtures prepared for 3D printing. Another follow-up paper from the same authors [19] revealed the effects of waiting time (between the time the mixture was prepared and the time of 3D printing using the mixture) on mechanical properties and rheological properties of the biomass–fungi mixtures, as well as the minimum printing pressure required for continuous extrusion during 3D printing, and the quality of the printed samples (characterized by layer-height shrinkage and filament-width uniformity). In another paper [13], researchers reported the development of a dedicated extrusion system for robotic arm printing. They also reported rheological and biological behaviors of the biomass–fungi materials. Other papers explored various aspects of 3D printing workflows, including the utilization of mycelium-based composite materials cultivated from waste sources [20], the printing of mycelium engineered living materials under non-sterile conditions [21], the effects of secondary colonizing time and substrate mixture on mechanical properties [22], and the effects of mixing parameters (mixing time and mixing mode) on fungal growth in biomass–fungi mixtures [23].

Mechanical properties (including strength) of biomass–fungi composite materials primarily rely on the binding of biomass particles by fungal hyphae. The feedstock material for the first process is a biomass–fungi material. To prepare this biomass–fungi material, biomass material is pasteurized at elevated temperatures to eliminate microorganisms that compete with fungi for resources to grow (such as nutrients). Then, fungi spores are introduced to the biomass material. The biomass–fungi material is subsequently dehydrated and packed into sterilized filter patch bags. In the primary colonizing process, water and wheat flour are added to the bags, creating a conducive environment for fungal growth for 4–5 days, resulting in foam-like biomass–fungi composite materials. Then, in the mixing process, these primary colonized biomass–fungi materials are mixed with additives like water, wheat flour, and psyllium husk powder to prepare the biomass–fungi mixture for 3D printing. Afterwards, in the 3D printing process, biomass–fungi mixture is used to print samples. After printing, in the secondary colonizing process, the printed samples are kept in a sterilized environment to allow further fungal growth in 3D printed samples. Finally, in the drying process, the 3D printed samples are dried at 90 °C, which is sufficiently high to kill all the fungi in the samples, ensuring that there will be no fungi growth under any humidity conditions.

After printing, the printed samples must go through secondary colonizing to facilitate fungal growth through the biomass particles and bind them. In some reported studies [12,13,20], significant height shrinkage and spreading of bottom layers were observed in 3D printed samples during the secondary colonization process, causing poor geometric accuracy of the 3D printed samples. However, there are no reported studies focusing on approaches to increasing geometric accuracy of the 3D printed samples from biomass–fungi composite materials.

This paper introduces a novel approach to enhance geometric accuracy of 3D printed samples by incorporating ionic crosslinking in the 3D printing-based methods. While ionic crosslinking has been widely utilized in bioprinting with hydrogel-based bioinks [24,25], its applications in biomass–fungi composite materials have not been reported. In many reported studies [26,27,28] on bioprinting with hydrogel-based bioinks and ionic crosslinking, sodium alginate (SA) was used as the hydrogel and calcium chloride (CaCl_2_) as the crosslinking agent. Crosslinking can enhance the mechanical properties of the 3D printed samples [27]. The authors have recently published an article regarding the effects of SA and CaCl_2_ on fungal growth and viability in 3D printing of biomass–fungi composite materials [29]. Another article showed ions could be used as regulators of fungal growth and development [30]. However, there are no reported studies regarding the effects of SA concentration and crosslinking exposure time on 3D printing of biomass–fungi composite materials. This paper aims to fill the gap by investigating the effects of SA on 3D printing and the effects of SA concentration and crosslinking exposure time on physiochemical characteristics of biomass–fungi mixture prepared for 3D printing and 3D printed samples.

## 2. Methodology

### 2.1. Procurement of Materials

The biomass–fungi material used in this study was sourced from GROW.bio (Green Island, NY, USA). It was supplied in a polypropylene bag with a square-shaped filter (1.5 × 1.5 inch, 0.2 µm pore size). The biomass material and the fungi strain were described in a previous article [12]. Wheat flour (all-purpose flour: Great Value, Walmart, Bentonville, AR, USA) and psyllium husk powder (NOW Supplements, Bloomingdale, IL, USA) were obtained from a local Walmart store. Sodium alginate (SA) (molecular weight 12,000 to 40,000 daltons) powder was procured from Milipore Sigma (Sigma-Aldrich, St. Louis, MO, USA). This SA powder composition was described in [31]. CaCl_2_ granules were purchased from Honeywell Electronic Chemicals (Charlotte, NC, USA).

### 2.2. Preparation of Biomass–Fungi Mixtures with Different SA Concentrations

#### 2.2.1. Preparation of Primary Colonized Material

Water and wheat flour were sterilized in an AMSCO 250LS autoclave (Mentor, OH, USA) at 120 °C for 50 min. Following sterilization, 700 mL of autoclaved water and 32 g of autoclaved wheat flour were added to the filter patch bag containing the as-received biomass–fungi material. The bag was shaken vigorously by hand for 1 min, then stored in a closet at 23 °C for five days to allow for primary colonization (i.e., primary colonizing) [18].

#### 2.2.2. Preparation of SA Solutions

SA solutions were prepared by following the procedure described in a previous paper [29]. SA solutions with three compositions were prepared: 0SA, 2SA, and 5SA solutions with an SA:water ratio of 0:100, 2:100, and 5:100 (g:mL), respectively. The 0SA solution was 100 mL of water (no SA was added).

#### 2.2.3. Preparation of Biomass–Fungi Mixtures with Different SA Concentrations

A mass of 50 g of primary colonized biomass–fungi material manually broken into small chunks, 100 mL of SA solution (0SA, 2SA, or 5SA solution), 100 mL of autoclaved water, and 20 g of autoclaved wheat flour were added in the mixer cup of a commercial mixer (NutriBullet PRO: Capital Brands, Los Angeles, CA, USA). The mixing parameters were set as follows: a mixing duration of 30 s and an intermittent mixing mode. Intermittent mixing involved mixing for five seconds, stopping, then manually shaking the mixer cup twice to ensure the mixture had good contact with the blades before resuming mixing. After mixing, 10 g of psyllium husk powder was added to the mixer cup and mixed into this mixture using a spatula. In this way, three biomass–fungi mixtures (0SA, 2SA, and 5SA biomass–fungi mixtures, respectively) with different SA concentrations were prepared.

### 2.3. Preparation of CaCl_2_ Crosslinking Agent

To prepare the crosslinking agent with the composition of 5:100 (5 g of CaCl_2_:100 mL of water), the procedure described in a previous paper was followed [29].

### 2.4. 3D Printing

The 3D printing experiments were conducted using a WASP Delta 2040 printer (WASP, Massa Lombarda, RA, Italy) equipped with an LDM (Liquid Deposition Modeling) extruder kit and a 3-L material storage container, as illustrated in Figure 2a. A pneumatically operated plastic piston facilitated the extrusion of the mixture through a custom-built nozzle assembly, which featured a screw extruder with a 6 mm opening (Figure 2b). The screw extruder, measuring 15 mm in diameter, 44 mm in height, and having an 11 mm pitch, rotated at 120 rpm, achieving a printing speed of 20 mm/s. All printing operations were performed at a room temperature of 23 °C.

The printer’s 3-L storage container was equipped with a plastic piston actuated by compressed air. The piston movement, driven by the printer’s motor, enabled the extrusion of the material through the nozzle assembly. The nozzle, guided by three stepper motors, moved along the *x*, *y*, and *z* axes, while the build platform remained stationary [12].

A 3D model of a cuboid (10 × 10 × 3 cm) (Figure 2c) was designed by Autodesk Fusion 360 (version 2.0.16985, Autodesk, San Francisco, CA, USA), then exported as an STL file. Slic3R (version 1.3.0.0) software was used to generate the G-code for printing. The samples were printed on a polypropylene film attached to the build platform, facilitating easy removal of the samples along with the film. The following printing parameters [19] were used: extrusion pressure = 2.5 bar, travel speed = 20 mm/s, layer height = 6 mm, and infill density = 30%.

### 2.5. Crosslinking

CaCl_2_ crosslinking agent was kept in a 15-quart (41.9 cm × 33 cm × 16.8 cm) plastic box (Sterilite Corporation, Townsend, MA, USA). The plastic box was half full of CaCl_2_ crosslinking agent. The 3D printed samples were fully submerged into the CaCl_2_ crosslinking agent for different lengths of exposure time. After crosslinking, the samples were kept directly on a polypropylene film (on the surface of a table) for 5 days at 23 °C. After 5 days, the 3D printed samples were dried for 4 h in a Black Decker extra wide convection oven (Black Decker, New Britain, CT, USA) set at 90 °C [12] to kill the fungi completely.

### 2.6. Assessment Methods for Quality of 3D Printed Samples

Quality of printed samples was assessed geometric accuracy and height shrinkage. Geometric accuracy was based on height and length deviation of the 3D printed samples with those of the model cuboid shape, as shown in Figure 2c. Height and length deviation were calculated following [32]. Height shrinkage on day 5 after printing was measured according to the procedure described in [19].

#### 2.6.1. Geometric Accuracy of 3D Printed Samples

Immediately after printing, digital photographs were taken from all the four sides of 3D printed samples using a Sony Alpha 9 II digital camera (Sony Corp., Minato, Tokyo, Japan). The length and height of the printed samples were measured using ImageJ (version 1.53s, National Institute of Health, Bethesda, MD, USA) software. Three length measurements were taken at each side (top, middle, and bottom, as illustrated in Figure 3) of the sample, and three height measurements were taken on each side (two corners and the middle, as illustrated in Figure 3) of the printed sample. Three samples were printed for each condition. For each sample, average length and height were calculated from the 12 length measurements and 12 height measurements on the sample, respectively. From this average length and height, length deviation and height deviation were calculated for each sample following [32]. For each condition, average length deviation (average of 3 sample length deviation) and average height deviation were calculated using the measurement data on the three samples printed under that condition and presented in this paper.

#### 2.6.2. Height Shrinkage of the 3D Printed Samples

To assess the height shrinkage of 3D printed samples during secondary colonization process, images of 3D printed samples were captured when they were freshly printed and on day 5 after printing (while going through secondary colonization). These images were analyzed using ImageJ software to obtain the height of the samples, following the methodology outlined in [19]. For each 3D printed sample, height was measured at three locations (two at the corners and one in the middle) on each side (a total of 12 height measurements). Three samples were printed for each condition. For each sample, average length and height were calculated from the 12 length measurements and 12 height measurements on the sample, respectively. For each condition, average height was calculated using the measurement data on the three samples printed under that condition and used in the calculation of height shrinkage. Height shrinkage on day 5 was calculated using the following equation:(1)$$Height\;shrinkage=\frac{H_{i}-H_{f}}{H_{i}}$$

Here, *H_i_* = average height of the printed samples when they were freshly printed and *H_f_* = average height of the printed samples on day 5.

### 2.7. Measurements of Rheological Properties of Biomass–Fungi Mixtures

A rheometer (ARES G2, TA Instruments, New Castle, DE, USA) was used to measure the rheological properties of biomass–fungi mixtures. For each viscosity measurement, 5 g of the control mixture (or 2SA biomass–fungi mixture or 5SA biomass–fungi mixture) was placed between parallel plates of 25 mm in diameter on the rheometer. First, viscosity was determined within the shear rate range from 0.1 to 100 s^−1^.

Second, oscillatory measurements were performed, with frequency increasing from 0.628 to 628 Hz, to evaluate the viscoelastic properties of a given material in response to the applied sinusoidal strain. The corresponding stress was measured. Elastic contribution is described by storage modulus (Pa), viscous contribution by loss modulus (Pa), and Tanδ value.

Finally, the three-step flow test [26] was performed to simulate the squeezing of the material through a nozzle. In step I, a shear rate of 0.1 s^−1^ was applied for 50 s. Then, in step II, a high shear rate of 50 s^−1^ was used for 10 s. In step III, a shear rate of 0.1 s^−1^ was applied for 240 s.

### 2.8. Measurements of Chemical Properties of 3D Printed Samples

Fourier transform infrared spectroscopy (FTIR) in Attenuated Total Reflection (ATR) mode was carried out in an iD7 ATR with Nicolet™ iS™ 5 Spectrometer (Thermo Fisher Scientific, Waltham, MA, USA) in the spectral range of 600–4000 cm^−1^ and resolution of 16 cm^−1^. The purpose was to identify the presence of ionic crosslinking in the 3D printed samples. To prepare the samples for FTIR measurements, a sharp scalpel was used to gently streak 5 g of mass from each 3D printed sample.

### 2.9. Measurements of Textural Properties of Biomass–Fungi Mixtures

Texture profile analysis (TPA) was performed on a texture analyzer (TA.XT.Plus, Texture Technologies, Hamilton, MA, USA). Cylindrical samples (TPA samples) were prepared using a cylindrical cup and a plunger following [19]. Three samples were prepared for each condition. The cup had an inner diameter of 15 mm and a height of 10 mm. The plunger had a diameter of 14.5 mm. The biomass–fungi mixtures with different SA concentrations were filled in the cup, then the plunger was used to ensure no air pockets in the samples. TPA was performed in triplicates for each condition.

During the measurement, TPA samples were compressed by a solid probe from an initial height of 10 mm to a final height of 2 mm (80% strain) at a constant rate of 0.5 mm/s (during the pre-test, test, and post-test), and a trigger force of 10.0 g was used [19]. The instrument began to record the data when the force reached the value of the trigger force. After the first compression (first downstroke) cycle, there was a delay of 5 s, then the TPA sample was compressed again (second downstroke). From each TPA test, a force–time (or force–distance) curve was obtained, as shown in Figure 4. Average hardness, springiness, and cohesiveness (from 3 TPA tests) were calculated following the procedure described in [19,33].

### 2.10. Statistical Analysis and Data Processing

Both *t*-test (one-tailed or two-tailed as necessary) and Tukey pairwise comparison were conducted using Minitab software (version 2019, Minitab Inc., State College, PA, USA) on the measurement results. Here, the *p*-value represents the lowest significance level at which the difference is statistically significant [34,35]. In other words, if the *p*-value obtained is higher than the preset significance level, the difference will not be statistically significant. The significance level in this study was set at 0.05. Origin Pro (version 2024b, OriginLab Corporation, Northampton, MA, USA) was used to prepare graphs and apply curve fittings.

## 3. Results and Discussion

### 3.1. Effects of SA Concentration and Crosslinking Exposure Time on Quality of 3D Printed Samples

#### 3.1.1. Geometric Accuracy of 3D Printed Samples

Table 1 presents dimensions and photographs of 3D printed samples using biomass–fungi mixtures with different SA concentrations and crosslinking exposure times. The 3D printed samples from the 0SA biomass–fungi mixture (control mixture) without crosslinking exhibited satisfactory geometric accuracy initially, albeit with occasional instances of over-extruded filaments (Figure 5a) in specific areas. However, over 5 days, noticeable discrepancies emerged: the bottom two layers of 3D printed samples widened compared with the fourth and fifth layers, with instances of collapse (Figure 5b) in certain areas. Consequently, these 3D printed samples had the poorest geometric accuracy compared with 3D printed samples from other biomass–fungi mixtures after 5 days.

The 3D printed samples from the 2SA biomass–fungi mixture with 1 min crosslinking exhibited satisfactory geometric accuracy initially. These samples maintained the best geometric accuracy after 5 days compared with other 3D printed samples. Conversely, 3D printed samples from the 2SA biomass–fungi mixture with 10 min crosslinking initially demonstrated good geometric accuracy; however, after 5 days, significant depressions (Figure 5c) were observed in some areas. Similarly, 3D printed samples from the 5SA biomass–fungi mixture with 1 min crosslinking displayed good printing accuracy initially, albeit with a lower initial height than 3D printed samples from the control, and 2SA biomass–fungi mixture with 1 min crosslinking. After 5 days, bending (Figure 5d) was observed in the corners of the 3D printed samples. Finally, 3D printed samples from the 5SA biomass–fungi mixture with 10 min crosslinking also exhibited good geometric accuracy initially. Still, they showed a lower initial height than 3D printed samples from both the control and 2SA biomass–fungi mixture with 1 min crosslinking. After 5 days, significant bending was observed in all the corners and significant depressions in certain areas of 3D printed samples from the 5SA biomass–fungi mixture with 10 min crosslinking. All the 3D printed samples from the 2SA and 5SA biomass–fungi mixtures performed better initially in geometric accuracy than 3D printed samples from the control mixture in maintaining geometric accuracy after 5 days.

#### 3.1.2. Height Shrinkage of 3D Printed Samples

Figure 6 shows the experimental data on height shrinkage of 3D printed samples from biomass–fungi mixtures with different SA concentrations and crosslinking exposure times. The error bars in the figure indicate 95% confidence intervals of the means. Firstly, 3D printed samples from the control mixture exhibited the highest height shrinkage on both day 3 and day 5 compared with the 3D printed samples from other mixtures. The 3D printed samples from 2SA biomass–fungi mixture with 1 min crosslinking exhibited the lowest height shrinkage on day 3 and day 5. The 3D printed samples from 2SA biomass–fungi mixture with 10 min crosslinking showed no significant difference (*p*-value ≥ 0.05, from individual two-tailed *t*-test) in height shrinkage compared with the 3D printed samples from 2SA biomass–fungi mixture with 1 min crosslinking. However, on day 5, 3D printed samples from 2SA biomass–fungi mixture with 10 min crosslinking exhibited significant depressions in some areas, resulting in a very high average and a wide confidence interval for height shrinkage compared with 3D printed samples from 2SA biomass–fungi mixture with 1 min crosslinking. Similarly, 3D printed samples from 5SA biomass–fungi mixture with either 1 min or 10 min crosslinking exhibited no significant difference in height shrinkage compared with 3D printed samples from 2SA biomass–fungi mixture with 1 min crosslinking on day 3 and day 5 (both *p*-values ≥ 0.05, from individual two-tailed *t*-test). However, significant depressions and bending were observed in the corners of all the 3D printed samples from 5SA biomass–fungi mixture with either 1 min or 10 min crosslinking. All the 3D printed samples from 2SA and 5SA biomass–fungi mixtures with different crosslinking exposure times exhibited significantly lower height shrinkage on day 5 compared with 3D printed samples from the control mixture (all the *p*-values < 0.05, from individual one-tailed *t*-test).

### 3.2. Effects of SA Concentration on Rheological Properties of Biomass–Fungi Mixtures

The printability of soft materials through extrusion is highly influenced by the rheological properties of the materials [36]. The viscosity of the material needs to be low enough at high shear rates to allow flow through a small nozzle, but the material has to quickly regain a high viscosity at rest to support the structure after deposition. However, viscosity measurements at high shear rates pose challenges for highly viscous materials using a rotational rheometer due to potential artifacts like shear fracturing and sample escape through the gap [37]. Hence, viscosity measurements were conducted within a low shear rate range of 0.1 to 100 s^−1^, closely mirroring the shear rates experienced during extrusion-based 3D printing [38].

Figure 7 illustrates that SA concentration had an impact on viscosity of the biomass–fungi mixture. The viscosity increased as SA concentration increased. A similar trend was observed by other researchers [39] when studying the effects of SA and rice variety on viscosity of the pastes. Both 2SA and 5SA biomass–fungi mixtures had higher viscosity than the control mixture. The prepared mixtures should display shear-thinning behavior to ensure continuous flow through the nozzle [40,41]. Remarkably, all biomass–fungi mixtures demonstrated pronounced shear-thinning behavior, with viscosity decreasing by several orders of magnitude as the shear rate increased from 0.1 to 100 s^−1^.

The power law model is widely used to represent the relationship between viscosity and shear rate [42]. A power law model is fitted using the viscosity data obtained from the measurements.
(2)$$\eta =K{\dot{\gamma }}^{n-1}$$
where n refers to the flow behavior index (dimensionless), $\dot{\gamma }$ indicates the shear rate (s^−1^), *η* means the viscosity of dispersions (Pa⋅s), and *K* denotes the consistency index (Pa⋅s^n^).

The values of *K* and n for the samples and R^2^ for the power law fitting are presented in Table 2. Viscosity (*η*)  values for all the samples at 10 s^−1^ are also included in Table 2. R^2^ values ranging from 0.9572 to 0.9985 suggest that the power law model is appropriate for representing the measured viscosity data at different shear rate values. Notably, different SA concentrations created differences in the flow consistency index and flow behavior index of the mixtures. The comparison of *K* at different n values is not ideal [43]; therefore, the viscosity (*η*) values at a shear rate of 10 s^−1^ were compared. Comparing with the control mixture, the *η* value at 10 s^−1^ was approximately six-fold higher for the 2SA biomass–fungi mixture, and about thirty-fold higher for the 5SA biomass–fungi mixture. The n values were consistently less than 1 for all the mixtures, representing their non-Newtonian fluid behavior. In general, a smaller *K* implies that less severe friction is required to extrude the mixture from the nozzle, which is beneficial for smooth extrusion [44]. Thus, the control mixture would face less friction compared with 2SA and 5SA biomass–fungi mixtures. This understanding can be used to explain the over-extrusion of 3D printed samples from the control mixture in certain areas, given that all the mixtures were printed under the same extrusion pressure and printing speed. The higher *K* value obtained for the 5SA biomass–fungi mixture might explain the lower initial height of the 3D printed samples as they faced more friction during extrusion than the other mixtures.

Viscoelastic properties of all the mixtures were also measured by oscillatory sweeps to determine whether these properties could be utilized to predict the printability of the materials. Figure 8 illustrates the storage modulus (G′) and loss modulus (G″) of the biomass–fungi mixtures with different SA concentrations at different angular frequencies. Storage modulus (G′) and loss modulus (G″) refer to the stored and dissipating energy during dynamic oscillations, respectively [45]. G′ exhibited elastic, solid-like behavior, while G″ was associated with viscous, liquid-like behavior [46,47]. It is noteworthy that, for all the mixtures, their G′ values were higher than their G″ values; and both G′ and G″ increased with angular frequency. This suggests that all these mixtures have a dominant elastic behavior, which is beneficial for better shape retention in 3D printed samples [48]. Despite the control mixture displaying the highest G′, printed samples from the control mixture had worse geometric accuracy than other mixtures. The 2SA and 5SA biomass–fungi mixtures had lower G′ but the printed samples from these mixtures demonstrated better geometric accuracy than the control mixture. Furthermore, the control mixture had the highest G″, possibly contributing to the spreading of bottom layers after printing subsequent layers.

Tanδ is defined as the ratio of loss modulus (G″) to storage modulus (G′). A material with Tanδ of less than 1 will show solid-like behavior; a material with Tanδ of larger than 1 will show predominantly liquid-like behavior [49]. The Tanδ values of all biomass–fungi mixtures with different SA concentrations are presented in Figure 8c. The 5SA biomass–fungi mixture had the lowest Tanδ value and the control mixture had the highest Tanδ value. In summary, the increase in SA concentration in the biomass–fungi mixture was associated with enhanced solid-like behavior.

As the 3D printed samples from both the 2SA and 5SA biomass–fungi mixtures demonstrated better print quality (geometric accuracy and height shrinkage) than those from the control mixture, these two mixtures were subjected to the three-step flow test [26]. This test showed the viscosity recovery values [26] of all mixtures by comparing the viscosity value at a shear rate of 0.1 s^−1^ during step III to the initial viscosity value measured during step I, which are before and after exposure to a high shear rate of 50 s^−1^ (step II) (Figure 9). Initially, the percentage of viscosity recovery before (at 30 s during step I) and after step II (at 60 s during step III) was compared. For the 5SA biomass–fungi mixture, approximately 100% of viscosity was recovered after step II, whereas for the 2SA biomass–fungi mixture, approximately 50% of viscosity was recovered. Again, the percentage of viscosity recovery before (at 30 s during step I) and after step II (at 284 s during step III) was compared. For the 5SA biomass–fungi mixture, only 31.27% of viscosity was recovered after step II, whereas for the 2SA biomass–fungi mixture, 42.25% of viscosity was recovered. The viscosity of the 5SA biomass–fungi mixture dropped rapidly with time whereas viscosity of 2SA biomass–fungi mixture did not drop significantly. The percentage of viscosity recovery decreased with increasing SA concentration. These findings imply that while higher SA concentrations may enhance certain rheological properties of the biomass–fungi mixture, they may not always lead to improved printing quality, which might also explain why the 3D printed samples from 2SA biomass–fungi mixtures had lower height shrinkage.

### 3.3. Effects of SA Concentration and Crosslinking Exposure Time on Chemical Properties of 3D Printed Samples

Fourier transform infrared (FTIR) spectroscopy was utilized to study the possible interactions between SA and crosslinking agent in FTIR samples extracted from the 3D printed samples from biomass–fungi mixtures with different SA concentrations and different crosslinking exposure times. This study did not include the control mixture because it did not contain any SA and did not have crosslinking.

Figure 10 shows the FTIR spectra results where spectrum “a” is obtained from FTIR sample of 2SA biomass–fungi mixture without crosslinking, spectrum “b” is obtained from FTIR sample of 2SA biomass–fungi mixture with 1 min crosslinking, spectrum “c” is obtained from FTIR sample of 2SA biomass–fungi mixture with 10 min crosslinking, spectrum “d” is obtained from FTIR sample of 5SA biomass–fungi mixture without crosslinking, spectrum “e” is obtained from FTIR sample of 5SA biomass–fungi mixture with 1 min crosslinking, and spectrum “f” is obtained from FTIR sample of 5SA biomass–fungi mixture with 10 min crosslinking. Across all FTIR samples, a prominent absorption band was observed within the 3600–3000 cm^−1^ range, attributable to the stretching vibration band of the −OH group [50].

FTIR spectrum of SA exhibited the bands around 3325, 1636, 1422, and 1025 cm^−1^, corresponding to the stretching vibrations of −OH, −COO− (asymmetric), −COO− (symmetric), and C−O−C, respectively [51]. The crosslinking process with Ca^2+^ ions caused a slight shift of the −COO− stretching bands to higher wavenumbers and a marginal reduction in the intensity (as evident in Figure 10, spectrum a vs. spectra b and c, and spectrum d vs. spectra e and f), indicating the formation of ionic bonding between Ca^2+^ and −COO− of SA. Additionally, the shoulder at 1080 ± 4 cm^−1^ relating to the C−O−C and C–O stretching can also be attributed to crosslinking [52,53].

### 3.4. Effects of SA Concentration and Crosslinking Exposure Time on Textural Properties of Biomass–Fungi Mixtures

Texture profile analysis (TPA) has been extensively used to characterize biological materials, including food materials (such as fruit, whipped toppings, gels, and pudding desserts) [54,55,56,57]. Textural properties (hardness, cohesiveness, and springiness) of biomass–fungi mixtures were measured by TPA.

Figure 11 shows the results obtained from TPA tests. The error bars in the figure indicate 95% confidence intervals of the means.

Hardness is the maximum force encountered during the first compression during TPA (Figure 4) [58,59]. The hardness value does not need to be at the point of maximum compression, although it typically does for most biological materials [60]. The TPA samples from 2SA biomass–fungi mixture without crosslinking did not show significant difference in hardness compared with TPA samples from the control mixture (*p*-value ≥ 0.05). However, TPA samples prepared from 5SA biomass–fungi mixture without crosslinking had a significantly higher hardness value than TPA samples prepared from the control mixture without crosslinking and TPA samples prepared from 2SA biomass–fungi mixture without crosslinking (both *p*-values < 0.05). TPA samples prepared from 2SA and 5SA biomass–fungi mixtures with different crosslinking exposure times exhibited significantly higher hardness values compared with TPA samples prepared from the control mixture (*p*-value < 0.05 for all). Thus, crosslinking exposure significantly increased hardness values of the TPA samples prepared from biomass–fungi mixtures with different SA concentrations. These results aligns with results obtained in another article [61] where the authors of that article had showed crosslinking typically resulted in a stiffer structure, as it increased the interaction between polymer chains, leading to enhanced rigidity and hardness. These hardness data align well with the experimental results regarding height shrinkage: height shrinkage was smaller for the 3D printed samples from 2SA and 5SA biomass–fungi mixtures with different crosslinking exposure times. When sodium alginate hydrogel is immersed in crosslinking agent for a longer time, the strength of the materials improves [61]. This increased strength was observed in the hardness measurements shown in Figure 11a. An increasing trend was observed in the hardness values when crosslinking exposure time was increased from 1 min to 10 min but the differences in hardness values were not significantly different. The authors will conduct a thorough study in the future to understand this phenomenon and its effect on height shrinkage of 3D printed biomass–fungi composite materials.

Cohesiveness represents how well the material withstands a second deformation relative to its resistance under the first deformation [62]. It is calculated by “the area of work” during the second compression (area 2 in Figure 4) divided by “the area of work” during the first compression (area 1 in Figure 4) [58,59]. An extruded layer of 3D printed part is considered cohesive if, after experiencing a compressive deformation, its internal structure is not so damaged that it cannot substantially resist a subsequent deformation (can be thought of as a successive layer of 3D printing). TPA samples prepared from both 2SA and 5SA biomass–fungi mixtures without crosslinking demonstrated significantly enhanced cohesiveness (*p*-value < 0.05) compared with TPA samples prepared from the control mixture (Figure 11b). The TPA samples exposed to crosslinking exhibited significantly reduced cohesiveness because they spread more during the first compression cycle, rather than remaining intact like less stiff, more cohesive materials. But, it would not impact 3D printing, as the crosslinking occurred post-printing.

Springiness represents how well a material physically springs back after it has been deformed during the first compression. It is measured during the second compression (note that there is a 5 s wait time between two compressions). The springiness is measured at the downstroke of the second compression [58,59]. In some cases, an excessively long wait time will allow a material to spring back more than it might under the conditions being researched (e.g., the wait time is less than 5 s instead of 60 s between subsequent layers of 3D printing). Thus, a 5 s wait time was considered for the TPA measurements. TPA samples prepared from both 2SA and 5SA biomass–fungi mixtures without crosslinking showed significantly higher springiness compared with TPA samples prepared from the control mixture (both *p*-values < 0.05), as shown in Figure 11c. The TPA samples were stiffer due to crosslinking exposure and this increased stiffness can also reduce the material’s ability to deform and recover its original shape after compression, leading to lower springiness.

Although all the samples exposed to crosslinking demonstrated significantly reduced springiness, it would not impact 3D printing since crosslinking occurred post-printing. This finding is consistent with that reported in the literature [63]. Mixtures with higher springiness indicated that the TPA sample was broken into a few large pieces during the first compression, whereas mixtures with lower springiness will result in TPA sample breaking into many small pieces. Mixtures with higher springiness demonstrated less spreading at the bottom layers of 3D printed samples (as observed in Table 1 for 3D printed samples from 2SA and 5SA biomass–fungi mixtures).

## 4. Conclusions

This paper presented a new approach to improve the quality of 3D printed samples (in terms of geometric accuracy and height shrinkage) by incorporating ionic crosslinking in 3D printing of biomass–fungi composite materials. It evaluated the effects of sodium alginate (SA) (hydrogel) and calcium chloride (crosslinking agent). Major conclusions are below.

Incorporating ionic crosslinking in 3D printing of biomass–fungi composite materials increased the geometrical accuracy and reduced the height shrinkage of printed samples. Notably, 3D printed samples from the 2SA biomass–fungi mixtures with 1 min crosslinking showed satisfactory geometric accuracy initially, maintained the best geometric accuracy and lowest height shrinkage even after 5 days, and did not exhibit significant depressions or bending.

Despite the control mixture exhibiting the highest storage modulus, it had the lowest loss modulus, resulting in the bottom layer spreading in the printed samples. Among the SA concentrations studied, as the content of SA increased, Tanδ values increased, and viscosity recovery decreased.

Fourier transform infrared (FTIR) spectroscopy confirmed ionic crosslinking in the 3D printed samples.

Furthermore, hardness values markedly improved with crosslinking exposure. The addition of sodium alginate to the biomass–fungi mixtures improved cohesiveness and springiness, enhancing the textural properties. This led to less spreading of the bottom layers in the 3D printed samples and minimized internal structure damage in the filaments.

## Figures and Tables

**Figure 1 biomimetics-09-00411-f001:**
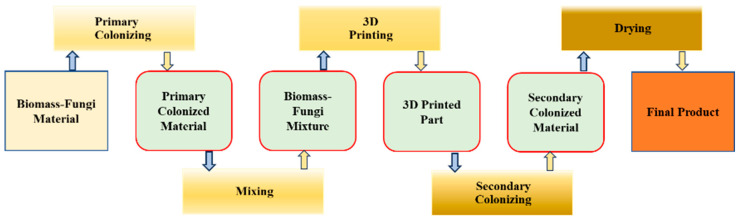
Processes of a 3D printing-based manufacturing method using biomass–fungi composite materials.

**Figure 2 biomimetics-09-00411-f002:**
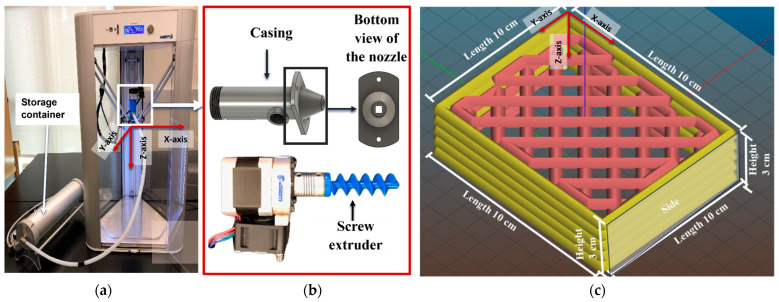
3D printing setup and sample design model: (**a**) 3D printer, (**b**) custom-built nozzle assembly, (**c**) model of the cuboid shape.

**Figure 3 biomimetics-09-00411-f003:**
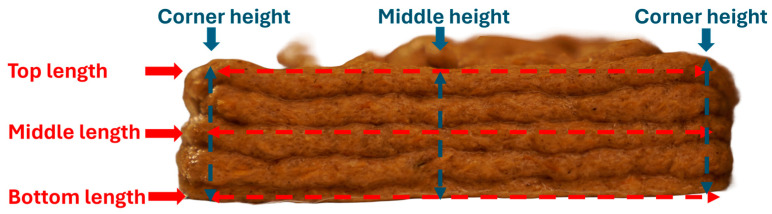
Location of measurements of 3D printed sample.

**Figure 4 biomimetics-09-00411-f004:**
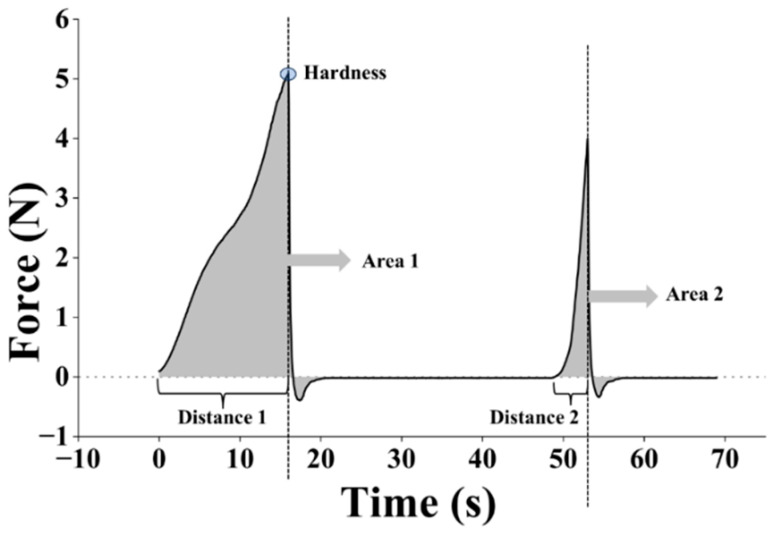
An example of force–time curve obtained from the texture analyzer.

**Figure 5 biomimetics-09-00411-f005:**
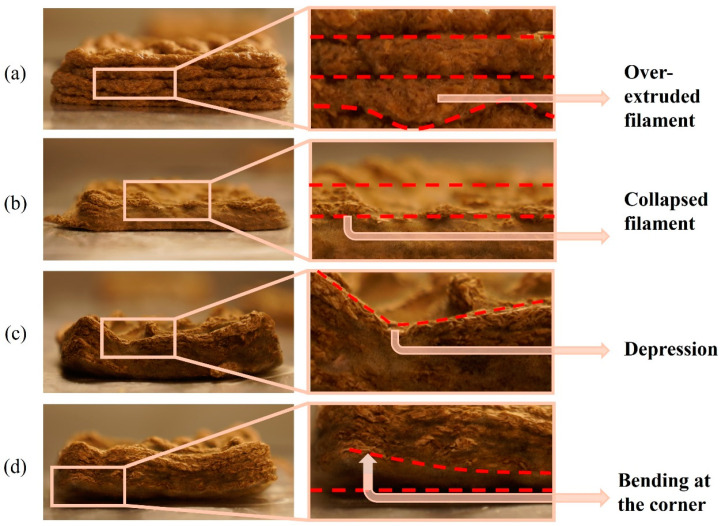
Discrepancies appeared in the 3D printed samples when they were freshly printed and day 5 after printing: (**a**) over-extruded filament in freshly 3D printed sample using 0SA biomass–fungi mixture, (**b**) collapsed filament after 5 days in 3D printed sample using 0SA biomass–fungi mixture, (**c**) depression after 5 days in 3D printed sample using 5SA biomass–fungi mixture, and (**d**) bending at the corner after 5 days in 3D printed sample using 5SA biomass–fungi mixture.

**Figure 6 biomimetics-09-00411-f006:**
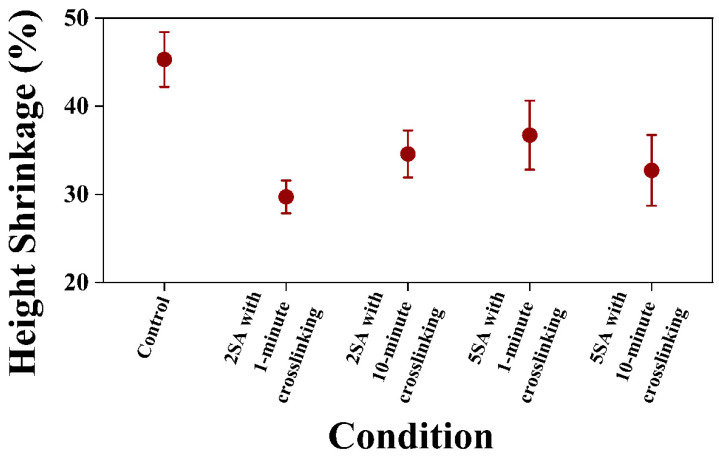
Height shrinkage on day 5 of 3D printed samples from biomass–fungi mixture with different SA concentrations and crosslinking exposure times.

**Figure 7 biomimetics-09-00411-f007:**
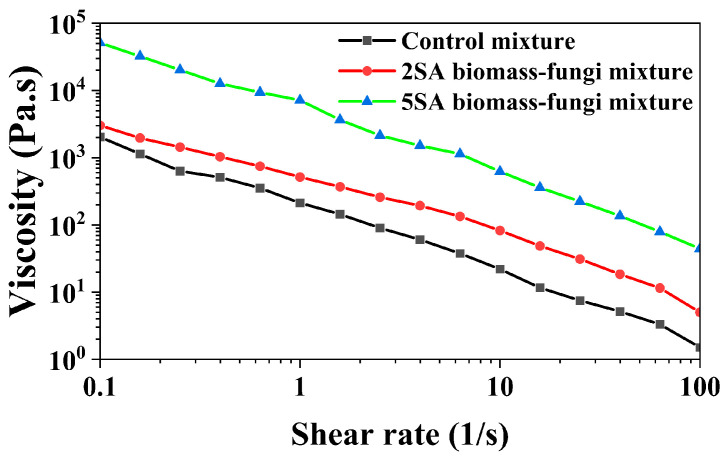
Viscosity versus shear rate from 0.1 to 100 s^− 1^ for biomass–fungi mixtures with different SA concentrations.

**Figure 8 biomimetics-09-00411-f008:**
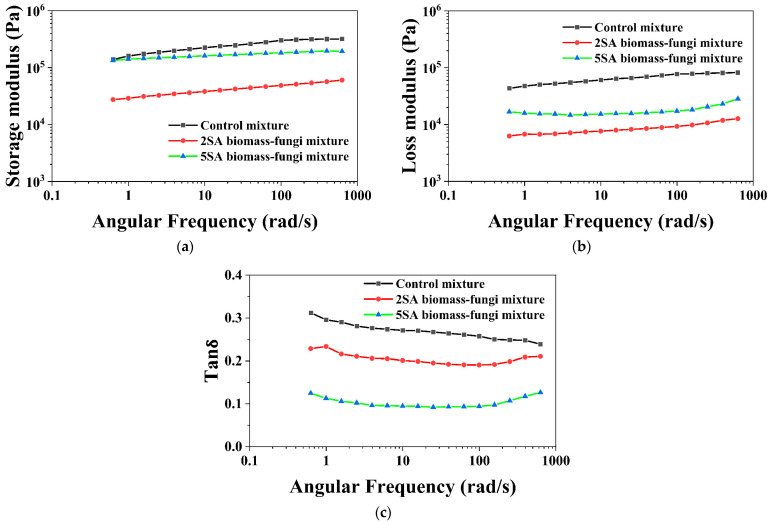
Viscoelastic properties of biomass–fungi mixtures with different SA concentrations: (**a**) storage modulus; (**b**) loss modulus; and (**c**) Tanδ.

**Figure 9 biomimetics-09-00411-f009:**
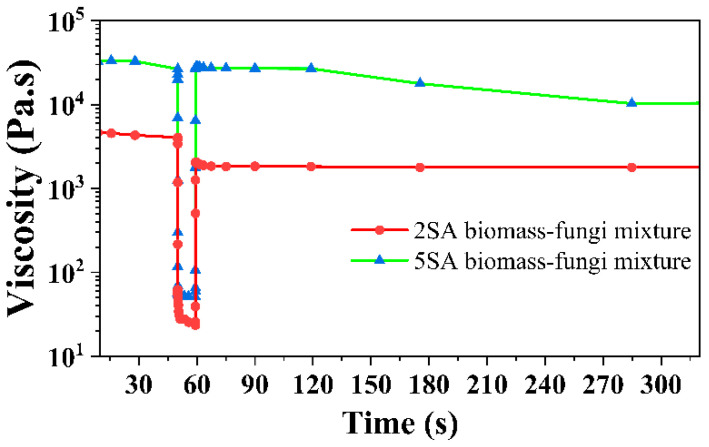
Viscosity versus time graph obtained as a result of the three step flow tests.

**Figure 10 biomimetics-09-00411-f010:**
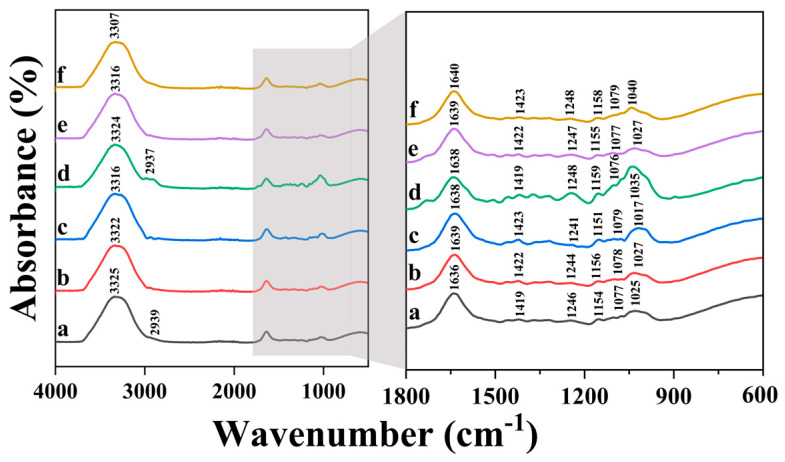
FTIR spectra: (**a**) 2SA biomass–fungi mixture without crosslinking, (**b**) 2SA biomass–fungi mixture with 1 min crosslinking, (**c**) 2SA biomass–fungi mixture with 10 min crosslinking, (**d**) 5SA biomass–fungi mixture without crosslinking, (**e**) 5SA biomass–fungi mixture with 1 min crosslinking, and (**f**) 5SA biomass–fungi mixture with 10 min crosslinking.

**Figure 11 biomimetics-09-00411-f011:**
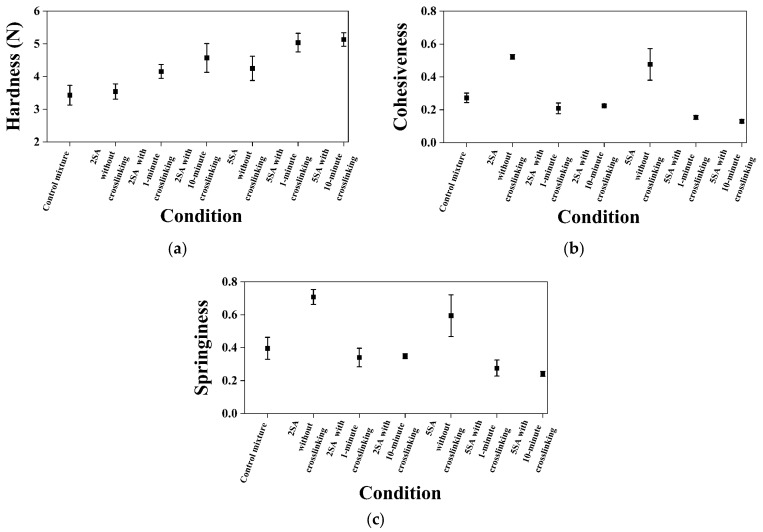
Texture profile analysis results for samples prepared from biomass–fungi mixtures with different SA concentrations and crosslinking exposure times: (**a**) hardness, (**b**) cohesiveness, (**c**) springiness.

**Table 1 biomimetics-09-00411-t001:** Dimensions and photographs of 3D printed samples with different SA concentrations and crosslinking exposure times.

Same Day after Printing				5 Days after Printing			
Condition	Photograph	Height Deviation (%)	Length Deviation (%)	Photograph	Height Deviation (%)	Length Deviation (%)	Observation
0SA biomass–fungi mixture (control mixture) without crosslinking	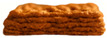	15.43 ± 5.48	2.72 ± 0.94	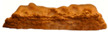	53.73 ± 17.04	7.44 ± 2.12	The bottom layers became wider as the subsequent layers were printed. After 5 days, the structure collapsed at some corners.
2SA biomass–fungi mixture with 1-min crosslinking	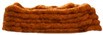	17.82 ± 1.96	3.12 ± 2.42	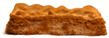	42.26 ± 11.28	16.2 ± 2.81	Good geometric accuracy, and no depressions observed.
2SA biomass–fungi mixture with 10-min crosslinking	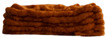	15.28 ± 1.17	5.45 ± 3.39	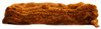	44.58 ± 10.77	18.32 ± 2.30	Good geometric accuracy initially, but after 5 days, depressions were observed in a few areas.
5SA biomass–fungi mixture with 1-min crosslinking	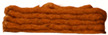	26.14 ± 6.92	5.02 ± 1.45	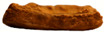	53.42 ± 12.24	9.44 ± 3.09	Good geometric accuracy initially, but after 5 days, the printed samples bent in some corners, and a few depressions were observed.
5SA biomass–fungi mixture with 10-min crosslinking	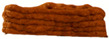	21.22 ± 2.42	10.41 ± 2.03	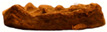	46.97 ± 13.68	15.18 ± 3.51	Good geometric accuracy initially, but after 5 days, significant depressions were observed, and printed samples bent in all the corners.

**Table 2 biomimetics-09-00411-t002:** Power law model parameters for viscosity of biomass–fungi mixtures with different SA concentrations.

Mixture	*K* (Pa·s)	*n*	R^2^	*η* (Pa·s) (at 10 s^−1^)
Control mixture	169	−0.066	0.9572	21.986
2SA Biomass–fungi mixture	2503	−0.172	0.9971	133.894
5SA Biomass–fungi mixture	5566	0.039	0.9985	623.684

## Data Availability

The authors confirm that the analyzed data that support the findings of this study are available within the article and others are available upon request.

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
