# Peer review of "Effects of Incorporating Ionic Crosslinking on 3D Printing of Biomass–Fungi Composite Materials"

_biomimetics, 2024, doi:10.3390/biomimetics9070411_

Round 1

Reviewer 1 Report

Comments and Suggestions for Authors

This manuscript investigates the effects of incorporating ionic crosslinking on the 3D printing of biomass-fungi composite materials. The authors vary the sodium alginate concentration in the printing ink and cure the printed samples in a calcium chloride solution to enable crosslinking for various durations, examining the geometry changes of the printed structures over 5 days. The authors also characterize the rheological properties, showing that introducing sodium alginate improves printability.

I have some major comments that may help improve the paper:

1.        In lines 25 and 104, the authors state that they use sodium alginate (SA) as the crosslinking agent and calcium chloride as the crosslinking solution. This statement is incorrect. Sodium alginate is a monomer/polymer, which requires calcium ions (Ca²) as the crosslinker to crosslink the material/hydrogel. Therefore, calcium chloride serves as the crosslinking agent, not sodium alginate. The authors should carefully review the reference cited in line 28 and other related works to correct this misunderstanding.

2.        In lines 58-72, the authors discuss the process of a 3D printing-based manufacturing method using biomass-fungi composite materials. Is this method the only method for 3D printing biomass-fungi composite material, or is it the one the authors used for their work? I suggest moving this entire paragraph to line 92 to improve the flow of the manuscript.

3.        The authors study the effect of 2SA and 5SA crosslinked for 1 and 5 minutes. What would the materials 2SA and 5SA be like without being placed in the crosslinking solution (meaning 0 minutes)? This is an important control group to demonstrate the effect of 1 minute of crosslinking. Additionally, the analysis of height changes, depressions, and bending does not seem like a thorough characterization method. It would be beneficial if the authors could develop a better quantification method for their future work.

4.        For rheology measurements, if the material has high viscosity, the authors might consider using a vane blade in the future instead of a round flat surface for measuring high shear rates.

5.        In line 413, measuring thixotropy does not usually run for a long time to compare the viscosity recovery rate. Typically, it would be around 30 seconds or immediately after the shear rate is reduced because this is directly related to the ability of the extruded material to retain its shape. If the authors calculate the value in a shorter time, then 5SA has a better recovery rate than 2SA. While it is interesting to note that the viscosity decreases for 5SA and not 2SA, potentially explaining the greater shrinkage of 5SA after 5 days, the material used for the rheology test is before the crosslinking solution. The authors should elaborate on this discussion a bit more.

Minor Comments:

There is a typo in line 160, and Figure 10 is cropped. The authors should carefully check the manuscript before resubmission.

Comments on the Quality of English Language

N/A

Author Response

The authors have responded to the reviewer's comments in the attached document.

Reviewer 2 Report

Comments and Suggestions for Authors

Page 4 line 146—what is autoclaved water? Water heated to 120C for 30 minutes? What is the purpose?

Section 2.4 is mistyped—says section 2.43. D printing and I assume should be section 2.4. 3D printing

Figure 5—labels of which is a fresh sample and which are 5 days should be listed in the caption as well as which is 0sa and 2sa

Section 3.1 lacks insight. For a lower SA concentration, longer crosslinking time seems to cause more shrinkage, but the same isn’t true for the higher SA concentration. What do we think the effect of a longer crosslinking time or higher SA concentration is on the material mechanistically? I would think that a longer crosslinking time would make the material more stable but that doesn’t seem to be the case.

Can we estimate the degree of crosslinking through viscosity measurements or make molecular weight measurements?

Line 379—why would increased friction result in a lower height? Is the overall volumetric flow rate lower when there is more friction? Was the nozzle pressure monitored during printing? I assume a higher nozzle pressure would be analogous to “higher friction”

Were the rheological measurements performed on samples that weren’t crosslinked with cacl2?

I think a better measurement than an oscillatory frequency sweep would have been a measurement of yield stress to predict yielding of the material after extrusion. Or some study of the relaxation behavior of the material may be more insightful as it seems the material is spreading and sinking over time, so some structural relaxation is taking place. I’m just not sure what the oscillatory measurements really accomplished or if any valuable conclusions could be drawn from them, especially if the samples weren’t crosslinked, as the relaxation and crosslinking should have been countering one another.

The drop in viscosity over time in Figure 9 of the 5SA sample is very interesting. Again it would be nice to see a further explanation of this or theories as to why this occurs in the presence of more SA. Is SA just plasticizing the material, or does it have a very short relaxation time?

In 3.5, again there is little discussion of the results or the implications of differences in the ftir spectra. I think it would be useful to see spectra of the biomass-fungi without SA and just the SA so we can have a better idea of which peaks are attributable to the different components and then what changes after crosslinking. Changes in peaks around 1422 and 1080 are said to be indicative of crosslinking, but there are notable changes in other peaks that are not discussed, and it is difficult to discern a trend.

Line 466—this is not a complete sentence

Why does crosslinking result in lower springiness and cohesiveness? I would imagine crosslinking would result in a stiffer structure, and I assume compressive or elastic modulus correlates with hardness. Are the samples actually broken during the first compression cycle, so the crosslinking samples break apart more because they are stiffer?

Comments on the Quality of English Language

Grammar was good, could benefit from minor edits 

Author Response

(The authors gave the same response as above.)

Round 2

Reviewer 1 Report

Comments and Suggestions for Authors

The authors have addressed all my comments and the manuscript is suggested to publish.

Reviewer 2 Report

Comments and Suggestions for Authors

thank you for your comprehensive answers